# Error rates of human reviewers during abstract screening in systematic reviews

Zhen Wang[1,2]*, Tarek Nayfeh[2], Jennifer Tetzlaff[3], Peter O'Blenis[3], Mohammad Hassan Murad[1,2]

1 Evidence-based Practice Center, Mayo Clinic, Rochester, Minnesota, United States of America, 2 Robert D. and Patricia E. Kern Center for the Science of Health Care Delivery Mayo Clinic, Rochester, Minnesota, United States of America, 3 Evidence Partners, Ottawa, Ontario, Canada

* Wang.Zhen@mayo.edu

## Abstract

### Background

Automated approaches to improve the efficiency of systematic reviews are greatly needed. When testing any of these approaches, the criterion standard of comparison (gold standard) is usually human reviewers. Yet, human reviewers make errors in inclusion and exclusion of references.

### Objectives

To determine citation false inclusion and false exclusion rates during abstract screening by pairs of independent reviewers. These rates can help in designing, testing and implementing automated approaches.

### Methods

We identified all systematic reviews conducted between 2010 and 2017 by an evidence-based practice center in the United States. Eligible reviews had to follow standard systematic review procedures with dual independent screening of abstracts and full texts, in which citation inclusion by one reviewer prompted automatic inclusion through the next level of screening. Disagreements between reviewers during full text screening were reconciled via consensus or arbitration by a third reviewer. A false inclusion or exclusion was defined as a decision made by a single reviewer that was inconsistent with the final included list of studies.

### Results

We analyzed a total of 139,467 citations that underwent 329,332 inclusion and exclusion decisions from 86 unique reviewers. The final systematic reviews included 5.48% of the potential references identified through bibliographic database search (95% confidence interval (CI): 2.38% to 8.58%). After abstract screening, the total error rate (false inclusion and false exclusion) was 10.76% (95% CI: 7.43% to 14.09%).

**Data Availability Statement:** All relevant data are within the manuscript and its Supporting Information files.

**Funding:** The author(s) received no specific funding for this work.

**Competing interests:** The authors have declared
that no competing interests exist.

## Conclusions

This study suggests important false inclusion and exclusion rates by human reviewers.
When deciding the validity of a future automated study selection algorithm, it is important to
keep in mind that the gold standard is not perfect and that achieving error rates similar to
humans may be adequate and can save resources and time.

## Introduction

Systematic review is a process to identify, select, synthesize and appraise all empirical evidence
that fits pre-specified criteria to answer a specific research question. Since Archie Cochrane
criticized lack of reliable evidence in medical care and called for "critical summary, by specialty
or subspecialty, adapted periodically, of all relevant randomized control trials" in 1970s [1],
systematic review has become the foundation of modern evidence based medicine. It is esti-
mated that the annual publications of systematic reviews increased 2,728% from 1,024 in 1991
to 28,959 in 2014.[2]

Despite of the surging number of published systematic reviews in recent years, many sys-
tematic reviews employ suboptimal methodological approaches.[2–4] Rigorous systematic
reviews require strict procedures with at least eight time-consuming steps.[5, 6] Significant
time and resources are needed, with estimated 0.9 minutes, 7 minutes and 53 minutes spent
per reference per reviewer on abstract screening, full text screening, and data extraction;
respectively.[7, 8] One thousand potential studies retrieved from literature search required 952
hours to complete.[9]Therefore, methods to improve efficiency of systematic reviews without
jeopardizing the validity are greatly needed.

In recent years, innovations have been proposed to accelerate the process of systematic
reviews, including methods to simplify steps of systematic reviews (e.g., rapid systematic
reviews)[10–13], and technology to facilitate literature retrieval, screening, and extraction.[7,
8, 14–21] Automation tools for systematic reviews, based on machine learning, text mining,
and natural language processing, have particularly been popular with an estimated workload
reduction from 30% to 70%.[14] Till July 2019, 39 tools have been completed and are available
for "real-world" use.[22] However, innovations are not always perfect and may introduce addi-
tional "unintended" errors. A recent study found an automation tool used by health systems to
identify patients with complex health needs led to significant racial bias.[23] Assessment of
these automation tools for systematic reviews, thus, is critical for wide adoption in practice.
[19, 24] No large scale test has been conducted. No conclusions have been made on whether
and how to implement these automation tools. Theoretically, assessment of the automation
tools can be treated as a classification problem: to determine whether a citation should be
included or excluded. The standard outcome metrics are used, such as sensitivity, specificity,
area under curve, positive predictive value. The standard of comparison (a.k.a., gold standard)
is usually human reviewers. Yet, human reviewers make errors. There is lack of evidence of
human errors in the process of systematic reviews.

Thus, we conducted this study to determine citation selection error rate (false inclusion and
false exclusion rates) in systematic reviews conducted by pairs of independent human review-
ers during abstract screening. These rates are currently unknown and can help in designing,
testing and implementing automated approaches.

## Materials and methods

### Study design and data source

We searched all systematic reviews conducted by an evidence-based practice center in the United States. The evidence-based practice center is one of the 12 evidence-based practice centers designated and funded by U.S. Agency for Healthcare Research and Quality (AHRQ). It specializes in conducting systematic reviews and meta-analysis, and developing clinical practice guidelines, evidence dissemination and implementation tools, and related methodological research. Eligible systematic reviews had to 1) be started and finished between June 1, 2010 and Dec 31, 2017; 2) follow standard systematic review procedures [5]: 1) dual independent screening of abstracts and titles, abstract inclusion by one reviewer prompted automatic inclusion for full text screening; 2) dual independent screening of full text, disagreements between reviewers reconciled via consensus or arbitration by a third reviewer. The final included list of studies consisted of the studies after abstract screening, and full texting screening.; 3) use a web-based commercial systematic review software (DistillerSR, Evidence Partners Incorporated, Ottawa, Canada); and 4) be led by at least one of the core investigators of the evidence-based practice center. The investigation team consisted of a core group (10–15 investigators at any time period) and external collaborators with either methodological or content expertise. DistillerSR was used to facilitate abstracts and full texts screening and track all inclusion and exclusion decisions made by human reviewers. We did not use any automation algorithm in the included systematic reviews.

### Outcomes

The main outcome of interest was error rate of human reviewers during abstract screening. An error was defined as a decision made by a single reviewer in abstract screening that was inconsistent (i.e., false inclusion or false exclusion) with the final included list of studies that that underwent abstract screening, and full texting screening and were eligible for data extraction and analysis (see Fig 1). We calculated error rate as the number of errors divided by the total number of screened abstracts (the total number of citations*2). We also estimated the overall abstract inclusion rate (defined as the number of eligible studies after abstract screening divided by the total number of citations), and the final inclusion rate (defined as the number of the final included studies divided by the total number of citations). In this study, we did not compare the performance between human reviewers and the automation algorithms integrated in DistillerSR.

### Statistical analysis

We calculated the outcomes of interest for each eligible systematic review. The mean of the outcomes across systematic reviews were the average outcomes of each study weighted by the inverse proportion to the variance of the denominator (total number of screened abstracts or total number of citations). The variance was estimated using the following formula:

$$V = \left\{ \frac{n}{[sum \ w_i(n-1)]} \right\} sum \ w_i(x_i - xbar)^2$$

$xbar$ = weight mean

All analyses were implemented with Stata version 15.1 (StataCorp LP, College Station, TX, USA).

## Results

A total of 25 systematic reviews were included in the analyses. These systematic reviews included 139,467 citations, representing 329,332 inclusion and exclusion decisions from 85

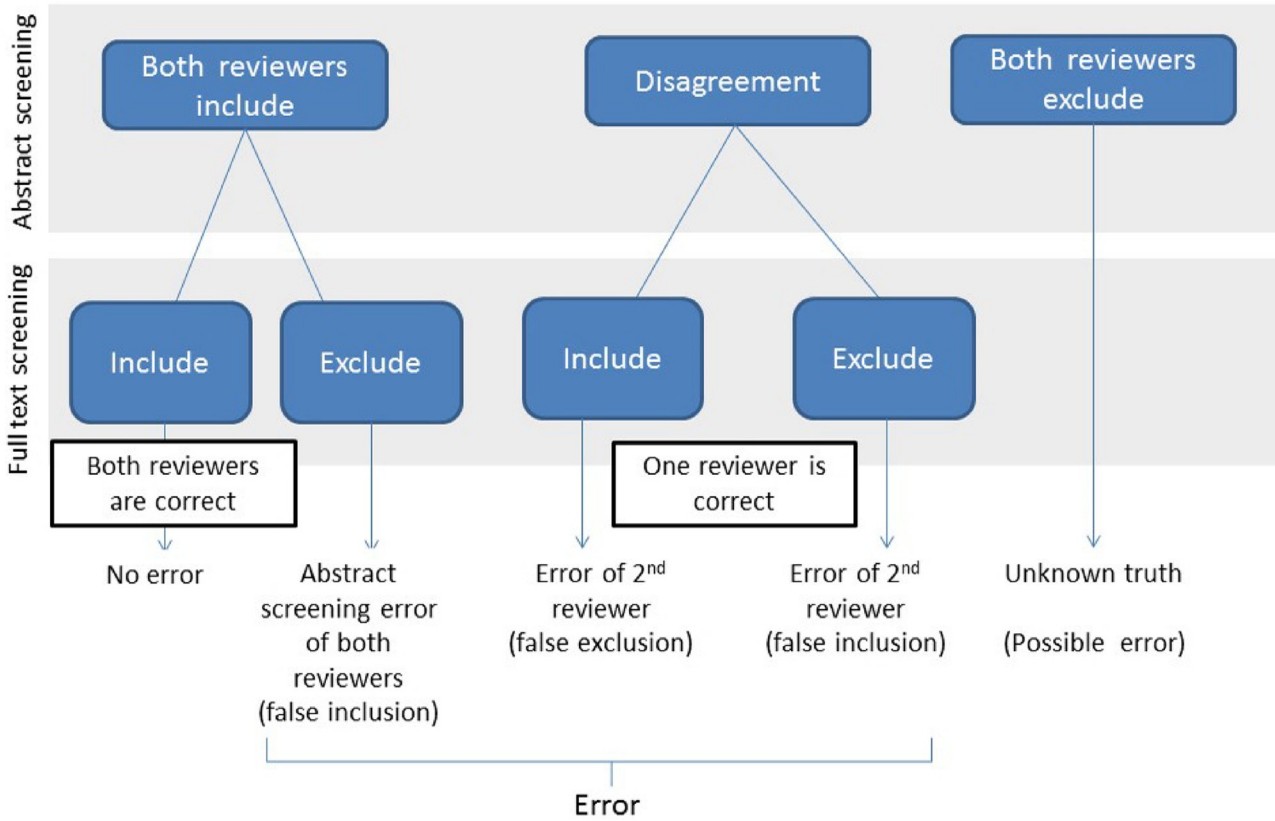

**Fig 1. Errors occurred during systematic review abstract screening.**

unique reviewers. Twenty-eight reviewers were core investigators from the evidence-based practice center; 57 were external collaborators with content or methodological expertise. Table 1 listed the characteristics of the included systematic reviews.

Abstract screening inclusion rate was 18.07% (95% CI: 12.65% to 23.48%) of the citations identified through literature search. Final inclusion rate after full text screening was 5.48% of the citations identified through literature search included in the systematic review (95% confidence interval (CI): 2.38% to 8.58%). The total error rate was 10.76% (95% CI: 7.43% to 14.09%). The error rates and inclusion rates varied by clinical area and type of review questions (Table 2).

## Discussion

In this cohort of 25 systematic reviews, covering 9 clinical areas and 3 types of clinical questions, a total of 329,332 screening decisions (inclusion vs. exclusion) were made by 85 human reviewers. The error rate (false inclusion and false exclusion) during abstract screening was 10.76%, which varied from 5.76% to 21.11%, depending on clinical areas and question types.

### Implications

A rigorous systematic review follows strict approaches and requires significant resource and time to complete, which typically lasts 6–18 months by a team of human reviewers.[25] Automation tools have the potential to mimic human activities in systematic review tasks and

**Table 1. Characteristics of the included systematic reviews.**

| Characteristics | Results |
|---|---|
| Systematic reviews | 25 |
| Time period | June 2010 to December 2017 |
| Citations from literature search | 139,467 |
| Inclusion and exclusion decisions | 329,332 |
| Decisions after abstract screening | 278,934 |
| Decisions after full text screening | 50,398 |
| Systematic reviewers | 85 |
| From the core team | 28 |
| External content or methodological experts | 57 |
| Clinical area | |
| Cardiovascular medicine | 1 |
| Mental health | 2 |
| Primary care | 3 |
| Pulmonology and critical care | 2 |
| Cardiovascular medicine | 1 |
| Endocrinology | 7 |
| Hematology | 2 |
| Health care delivery research | 4 |
| Urology | 3 |
| Review question type | |
| Methodology | 4 |
| Diagnostic/Screening/Prognostic | 4 |
| Treatment | 17 |

gained popularity in academia and industry. However, validity of the automation tools has yet to be established. [19, 24] It is intuitive to assume that these tools should achieve a zero error rate in order to be implemented to generate evidence used for decision-making (i.e., 100% sensitivity and 100% specificity).

Human reviewers have been used as the "gold standard" in evaluating the automation tools. However, similar to those "gold standards" used in clinical medicine, 100% accuracy is unlikely in reality. We found 10.76% error rate made by human reviewers in abstract screening (an error about 1 in 9 abstracts). This error rate also varied from topics and types of questions. Thus, when developing and refining an automation tool, achieving error rates similar to humans may be adequate. If this is the case, then these tools can serve as a single reviewer that gets paired with a second human reviewer.

## Limitations

The sample size is relatively small, especially as we further stratify by clinical areas. The findings may not be generalizable to other systematic review questions or topics. The human reviewers who conducted these systematic reviews had a wide range of content knowledge and methodological experience (from minimum 1 year to over 10 years), which can be quite different from other review teams. In our practice, citations from abstract screening were automatically included when conflicts between two independent reviewers emerged. The abstract inclusion rate and final inclusion rate resulting from this approach can be higher than those of the teams who resolve conflicts in abstract screening. When both reviewers agree on excluding an abstract, this abstract disappears from the process; thus, a dual erroneous exclusion cannot

**Table 2. Error and inclusion rates by topic area and type of review questions.**

|  | Final inclusion rate (95% CI) dual process | Abstract inclusion rate (95% CI) dual process | Error rate (95% CI) |
|---|---|---|---|
| **Overall (n = 25)** | 5.48% (2.38% to 8.58%) | 18.07% (12.65% to 23.48%) | 10.76% (7.43% to 14.09%) |
| **Clinical Area** |  |  |  |
| **Cardiovascular medicine (n = 1)** | 1.59% | 23.92% | 17.73% |
| **Mental health (n = 2)** | 2.10% (0% to 20.15%) | 9.92% (0.77% to19.06%) | 6.43% (0% to 18.19%) |
| **Primary care (n = 3)** | 5.39% (0% to 13.19%) | 28.00% (20.85%, 35.15%) | 21.11% (5.15% to 37.08%) |
| **Pulmonology and critical care (n = 2)** | 1.12% (0% to 2.43%) | 9.13% (05 to 38.18%) | 6.68% (0% to 42.04%) |
| **Cardiovascular medicine (n = 1)** | 1.93% | 18.56% | 19.16% |
| **Endocrinology (n = 7)** | 5.91% (2.89% to 8.94%) | 20.40% (9.70% to 31.09%) | 12.23% (4.76% to 19.70%) |
| **Hematology (n = 2)** | 6.69% (0% to 37.32%) | 11.00% (0% to 40.13%) | 5.76% (0% to 24.27%) |
| **Health care delivery research (n = 4)** | 2.18% (0% to 6.86%) | 14.55% (0% to 43.35%) | 8.73% (0% to 29.03%) |
| **Urology (n = 3)** | 23.77% (0% to 57.85%) | 42.85% (24.92% to 60.79%) | 17.17% (0% to 36.07%) |
| **Review question type** |  |  |  |
| **Methodology** | 2.18% (0.00% to 6.86%) | 14.55% (0% to 43.35%) | 8.73% (0.00% to 29.03% |
| **Diagnostic/Screening/Prognostic** | 7.83% (3.56% to 12.11%) | 25.99% (0% to 57.82%) | 14.97% (0% to 34.38%) |
| **Treatment** | 5.86% (1.47% to 10.26%) | 17.64% (12.00% to 23.29%) | 10.57% (7.32% to 13.83%) |

be assessed. We were not able to evaluate error rate during full text screening as we did not track the conflicts between reviewers. Lastly, while we call judgments in study selection that are inconsistent with the final inclusion as errors, we acknowledge that these errors could be due to poor reporting and insufficient data provided in the published abstract. Thus, they may not be avoidable and they are not the fault of human reviewers. In summary, this study is an initial step to evaluate human errors in systematic reviews. Future studies need to evaluate different systematic review approaches (e.g., rapid systematic review, scoping review), clinical areas, and review questions. It is also important to increase the number of systematic reviews involved in the evaluation and include other EPC or non-EPC institutions.

## Conclusions

This study of 329,332 abstract screening decisions made by a large, diverse group of systematic reviewers suggests important false inclusion and exclusion rates by human reviewers. When deciding the validity of a future automated study selection algorithm, it is important to keep in mind that the gold standard is not perfect and that achieving error rates similar to humans is likely adequate and can save resources and time.

## Supporting information

**S1 Appendix.**
(DOCX)

## Author Contributions

**Conceptualization:** Zhen Wang, Jennifer Tetzlaff, Peter O'Blenis, Mohammad Hassan Murad.

**Data curation:** Zhen Wang.

**Formal analysis:** Zhen Wang, Tarek Nayfeh, Mohammad Hassan Murad.

**Investigation:** Zhen Wang.

**Methodology:** Zhen Wang, Jennifer Tetzlaff.

**Project administration:** Zhen Wang, Tarek Nayfeh.

**Supervision:** Zhen Wang, Peter O'Blenis, Mohammad Hassan Murad.

**Validation:** Zhen Wang.

**Visualization:** Zhen Wang.

**Writing – original draft:** Zhen Wang, Mohammad Hassan Murad.

**Writing – review & editing:** Zhen Wang, Tarek Nayfeh, Jennifer Tetzlaff, Peter O'Blenis, Mohammad Hassan Murad.

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
