## [Decision Letter · Decision Letter 0]

12 Dec 2019

PONE-D-19-26633

Error rates of human reviewers during abstract screening in systematic reviews

PLOS ONE

Dear Dr. Wang,

Thank you for submitting your manuscript to PLOS ONE. After careful consideration, we feel that it has merit but does not fully meet PLOS ONE’s publication criteria as it currently stands. Therefore, we invite you to submit a revised version of the manuscript that addresses the points raised during the review process.

This is an interesting study. However, as pointed out by two reviewers, there are some issues needed to be addressed in particular the methodology. Since the work is quite unique, more explanation in the background (see Reviewer #1) as well as the Methods (see Reviewers #1 and 3), are needed. 

We would appreciate receiving your revised manuscript by Jan 26 2020 11:59PM. To enhance the reproducibility of your results, we recommend that if applicable you deposit your laboratory protocols in protocols.io, where a protocol can be assigned its own identifier (DOI) such that it can be cited independently in the future. For instructions see: http://journals.plos.org/plosone/s/submission-guidelines#loc-laboratory-protocols

We look forward to receiving your revised manuscript.

Kind regards,

Sompop Bencharit, DDS, MS, PhD, FACP

Academic Editor

PLOS ONE

Journal Requirements:

1. Please ensure that you include a title page within your main document. You should list all authors and all affiliations as per our author instructions and clearly indicate the corresponding author.

2. Please upload a copy of Figure, to which you refer in your text on page 3. If the figure is no longer to be included as part of the submission please remove all reference to it within the text.

Reviewers' comments:

Reviewer's Responses to Questions

**Comments to the Author**

1. Is the manuscript technically sound, and do the data support the conclusions?

Reviewer #1: Yes

Reviewer #2: Yes

Reviewer #3: No

2. Has the statistical analysis been performed appropriately and rigorously? 

Reviewer #1: Yes

Reviewer #2: I Don't Know

Reviewer #3: No

3. Have the authors made all data underlying the findings in their manuscript fully available?

Reviewer #1: No

Reviewer #2: Yes

Reviewer #3: Yes

4. Is the manuscript presented in an intelligible fashion and written in standard English?

Reviewer #1: Yes

Reviewer #2: Yes

Reviewer #3: Yes

5. Review Comments to the Author

Reviewer #1: Thank you for this study which contributes to our knowledge of the process of conducting systematic reviews.

Data Availability: The list of systematic reviews examined is provided; however, there are no data regarding the full sets of references obtained from the original searches of the examined systematic reviews, nor the inclusion/exclusion decisions of each reviewer for each systematic review. Therefore, readers would not be able to replicate this analysis with the data currently provided.

Statistical Analysis: the descriptive statistics were calculated correctly.

Background: You mention that systematic reviews employ suboptimal methodological approaches and the potential for human errors; however, you do not acknowledge that automated systems and algorithms could introduce errors, potentially systematic errors which could introduce bias into a systematic review, such as that found in this study https://science.sciencemag.org/content/366/6464/447

Methods: Consider explaining/justifying the eligibility criteria #3 use a single software program, Distiller SR - were reviews excluded because they used a different software program? For eligibility criteria #2 - was the citation inclusion by one reviewer prompting automatic inclusion happen at only the abstract screening level (vs. full text screening)? under what circumstances were disagreements between reviewers reconciled via consensus or arbitration.

Results: Consider breaking out the error rate to errors of exclusion and errors of inclusion, as these may be different and of interest to readers and would allow you to provide rates of specificity and sensitivity, typical metrics for classification problems as you mention in the Background section and as was done in the study by Bannach-Brown referenced below.

Further, I would like to see a breakdown of errors from the abstract screening vs. the full text screening (and within this stratification, report exclusion vs. inclusion errors). This is important and interesting, because as you note in the limitations, the errors of inclusion at the abstract screening level reflect the fact that more information is needed to make a decision, rather than commission of an error by the reviewer.

Limitations: You note that "In our practice, citations from abstract screening were automatically included when conflicts between two independent reviewers emerged" - perhaps note that this may have resulted in a spuriously lower error rate than "truth."

Consider referencing the following article in the background or implications as this study calculated error rates from a Machine Learning screening algorithm: Bannach-Brown, A., Przybyła, P., Thomas, J., Rice, A. S., Ananiadou, S., Liao, J., & Macleod, M. R. (2019). Machine learning algorithms for systematic review: reducing workload in a preclinical review of animal studies and reducing human screening error. Systematic reviews, 8(1), 23.

Limitations or perhaps conclusions: As this is a novel study, consider contextualizing the findings as an initial calculation of human errors observed in a small number of systematic reviews and discussing the need for replication of this study using reviews conducted by other EPCs or other non-EPC institutions, using different software, and most important to increase the sample size of studies used to inform our knowledge of valid human error rates.

Typo: 3rd paragraph of background section, In recent year should be In recent years

Reviewer #2: Dear authors - Thank you for investigating this question. It's of interest to me.

I agree that the sample is small and I wondered how meaningful the 10% error rate is. It's expected that human reviewers will make mistakes and if each reviewer errors on 10% of decisions, then... what? That's one reason why there are two reviewers. However, the use of that number as a benchmark to test automated systems puts it in an interesting context. If the automated process is as effective as a human reviewer, then we can save significant human time by eliminating one of the reviewers. But, again, the sample is small, and I imagine the type of question can influence the error rate, and some screening questions might be more conducive to human review / automation. In other words, a 10% error rate for one question may vary drastically for another. Some of this is touched on in the 'implications' section, which I'd like to see expanded, but I understand that such discussion can easily go beyond the study.

Overall, the small sample and the topic variability in the sample and the variability of the screening questions in each of the reviews in the sample lead me to question the significance of the 10%. And that makes me question its utility as a benchmark. But it's a thought-provoking topic and the authors use an interesting method to address it (Distiller data).

Thank you, by the way, for including the complete list of systematic reviews included in the analysis. This helps with reproducibility.

Reviewer #3: Dear Authors,

I have attached a revised version of your manuscript and a PDF file including my specific comments. My major concerns are related to the methdology you followed in your study.

Please, check both.

Best regards.

6. PLOS authors have the option to publish the peer review history of their article (what does this mean?). If published, this will include your full peer review and any attached files.

Reviewer #1: No

Reviewer #2: Yes: Mark MacEachern

Reviewer #3: No

---

## [Author Response · Author response to Decision Letter 0]

24 Dec 2019

Please see our responses in the Response to Reviewers.docx

---

## [Editor Report · Decision Letter 1]

30 Dec 2019

Error rates of human reviewers during abstract screening in systematic reviews

PONE-D-19-26633R1

Dear Dr. Wang,

We are pleased to inform you that your manuscript has been judged scientifically suitable for publication and will be formally accepted for publication once it complies with all outstanding technical requirements.

With kind regards,

Sompop Bencharit, DDS, MS, PhD, FACP

Academic Editor

PLOS ONE

Additional Editor Comments (optional):

The authors have sufficiently addressed all comments from the reviewers.
---

## [Editor Report · Acceptance letter]

2 Jan 2020

PONE-D-19-26633R1 

Error rates of human reviewers during abstract screening in systematic reviews 

Dear Dr. Wang:

I am pleased to inform you that your manuscript has been deemed suitable for publication in PLOS ONE. Congratulations! Your manuscript is now with our production department. 

With kind regards,

on behalf of

Dr. Sompop Bencharit 

Academic Editor

PLOS ONE